**PLOS** NEGLECTED TROPICAL DISEASES

# Modelled impact of Tiny Targets on the distribution and abundance of riverine tsetse

**Glyn A. Vale**[1,2]*, **John W. Hargrove**[1], **Andrew Hope**[3], **Steve J. Torr** [3]*

**1** South African Centre for Epidemiological Modelling and Analysis, University of Stellenbosch, Stellenbosch, South Africa, **2** Natural Resources Institute, University of Greenwich, Chatham, United Kingdom, **3** Liverpool School of Tropical Medicine, Liverpool, Merseyside, United Kingdom

* valeglyn@gmail.com (GAV); steve.torr@lstmed.ac.uk (SJT)

## Abstract

### Background

The insecticide-treated baits known as Tiny Targets are one of the cheapest means of controlling riverine species of tsetse flies, the vectors of the trypanosomes that cause sleeping sickness in humans. Models of the efficacy of these targets deployed near rivers are potentially useful in planning control campaigns and highlighting the principles involved.

### Methods and principal findings

To evaluate the potential of models, we produced a simple non-seasonal model of the births, deaths, mobility and aging of tsetse, and we programmed it to simulate the impact of seven years of target use against the tsetse, *Glossina fuscipes fuscipes*, in the riverine habitats of NW Uganda. Particular attention was given to demonstrating that the model could explain three matters of interest: (i) good control can be achieved despite the degradation of targets, (ii) local elimination of tsetse is impossible if invasion sources are not tackled, and (iii) with invasion and target degradation it is difficult to detect any effect of control on the age structure of the tsetse population.

### Conclusions

Despite its simplifications, the model can assist planning and teaching, but allowance should be made for any complications due to seasonality and management challenges associated with greater scale.

## Author summary

We produced a simple model of the population dynamics of the riverine tsetse fly, *Glossina fuscipes fuscipes*, to simulate the field results of controlling this insect for seven years using Tiny Targets, i.e., artificial insecticide-treated baits, in NW Uganda. The model is potentially useful in planning tsetse control and teaching the principles involved. Thus, it confirmed that targets can give good control, even if the targets degrade so much that they

**Data Availability Statement:** All relevant data are within the manuscript and its Supporting Information files. The programme (Tsetse Plan 2)

is publicly available from SourceForge with the URL: https://sourceforge.net/projects/tsetse-plan-2/.

**Funding:** Funding for this research was provided by the Bill and Melinda Gates Foundation (www.gatesfoundation.org), with grants awarded to SJT (Grant ID#: OPP1104516, INV-001785), and the UK Biotechnology and Biological Sciences Research Council with grants awarded to SJT (BB/S01375X/1, BB/L019035/1). SACEMA (GAV, JWH) receives core funding from the Department of Science and Innovation, Government of South Africa. The funders had no role in study design, data collection and analysis, decision to publish, or preparation of the manuscript.

**Competing interests:** The authors have declared that no competing interests exist.

need replacement after six months. It showed that invasion can limit severely the efficacy of control campaigns and mask the changes in the age structure of the population. We stress the need to consider the possibility of seasonal complications and problems of managing large scale operations.

## Introduction

Tsetse flies (*Glossina* spp) transmit the subspecies of *Trypanosoma brucei* that cause human African trypanosomiasis (HAT), commonly called sleeping sickness. There are two forms of this disease. The most common form is Gambian HAT (gHAT), caused by *T. b. gambiense* transmitted by riverine species of tsetse. The rarer Rhodesian HAT (rHAT) is caused by *T. b. rhodesiense*, which is usually spread by savanna species of tsetse. The distribution and incidence of HAT is shrinking, due to case detection and treatment, coupled with vector control [1]. The WHO aims to eliminate the transmission of sleeping sickness across Africa by 2030 [2], and elimination as a public health problem has already been declared in Benin, Cote d'Ivoire, Equatorial Guinea, Rwanda, Togo, and Uganda [3].

For gHAT, the most common method of vector control is to deploy artificial insecticide-treated baits, known as Tiny Targets, in the riverine vegetation where the vectors concentrate [4–6]. The ability to restrict the deployments to such vegetation means that humans can often be protected by killing the flies in comparatively small areas. This contrasts with the fact that the usual savanna vectors of rHAT must be controlled evenly over very extensive areas. However, getting the best from the Tiny Targets in riverine situations involves balancing the various opposing considerations within each of the following two spheres.

First, riverine tsetse can have a mean net movement of ~300 m/day if the habitat is sufficiently extensive [7]. This high mobility is advantageous, from the point of view of tsetse control, in that it allows the flies to locate the stationary targets. However, it can also be disadvantageous in that it provides a stream of flies invading from uncontrolled areas nearby. Such invasion means that it is impossible to ensure complete elimination of tsetse unless the whole extent of the infested area is tackled, up to the natural limits to invasion. Some mitigation of this problem arises because controlling tsetse by only 60–90% can be sufficient to break the transmission cycle of gHAT and so eliminate the disease locally [8, 9].

Second, Tiny Targets are simple, relatively cheap [10–12], and highly effective provided the targets remain in place and remain effective. Unfortunately, some of the targets can be washed away by floods or destroyed by bushfires, and others can be stolen or damaged by people [5]. Moreover, the efficacy of the insecticide deposit degrades over several months [5]. Against these problems, the fact that a mere 60–90% control of tsetse can be adequate to stop disease transmission suggests that the target technology can tolerate substantial misuse.

Models that address the above considerations would assist the planning and management of control campaigns by predicting the level and distribution of tsetse control achieved by different ways of using targets. Moreover, by highlighting various aspects of population dynamics, models are useful aids to teaching and research planning. The most pertinent model in previous use is Tsetse Muse (TM) [13], but its spatial simulations are primarily one dimensional. Present work developed TM into a new, two-dimensional model, termed Tsetse Plan 2 (TP2), that can deal with a wider range of operational areas, including those comprising a network of river lines. This new model was then employed to simulate and explain the field experience of the use of Tiny Targets to control the riverine tsetse, *G. fuscipes fuscipes*, in the focus of gHAT in NW Uganda [5].

We used the model to answer three research questions. First, why was the impact of Tiny Targets greater in the upstream sections of rivers and streams? Second, while the targets rapidly reduced the abundance of tsetse to low levels, why did they not eliminate the tsetse population? Third, why did the targets not produce a noticeable reduction in the mean age of the population. We hypothesized that reductions in the anticipated levels of control were consequences of (i) natural movement of tsetse, (ii) variation in the abundance of tsetse habitat along the riverbanks and (iii) the loss and degradation of targets following their deployment. In addition to answering these questions, we also used the model to assess the rate and extent to which tsetse populations would recover following the cessation of vector control. The motivation for this last investigation was the expectation that tsetse control would be scaled back when gHAT was eliminated. In all the above modelling there was the problem that the values of many of the necessary input parameters were unreliably quantified, so we explored the effects of a range of values.

## Methods

### Model structure

The model is downloadable for inspection and use at https://sourceforge.net/projects/tsetse-plan-2/, and is merely outlined here. It was produced in spreadsheets of Microsoft Excel and was operated via Visual Basic for Applications. The spreadsheets represented lifetables with compartments for each daily age class of adult male and female tsetse and pupae. Each compartment took the form of a map, comprising a block of 70 x 70 cells, each representing 1 x 1km of territory, so that the whole map covered 70 x 70km. Numbers in each cell of the map indicated the distribution of the age class to which the compartment referred. The central 50 x 50 cells of the map were intended to portray the block of roughly 50 x 50km studied by Hope et al. in NW Uganda [5]. The band of cells surrounding the central block represented the potential invasion sources, comprising the nearby parts of Uganda to the North, East and South, and the adjacent part of the Democratic Republic of Congo (DRC) to the West. The border area between the countries comprises a watershed running North to South with relatively sparse vegetation. On the Ugandan side of the watershed the rivers drain East to the Nile. On the DRC side, the rivers go West to the Atlantic.

Detailed portrayal of the varied vegetation of the mapped area was beyond the scope of present work. Hence, the vegetation map was produced by combining just four types of cell, regarded as being traversed by: (1) a large river, (2) medium river, (3) small river, and (4) no river, i.e., an interfluve. Unless stated otherwise, the percent of each sort of cell that was covered by good habitat was 10%, 7%, 4% and 1%, respectively, and for cells with a river the habitat was considered to occur along the river. For example, the 10% of habitat cover in cells with a large river would involve bands of habitat averaging 50m wide along each bank, assuming that the river ran straight through the cell. Within those cells that contained a river, the vegetation away from the riverine habitat was regarded as the same as in the interfluve. Fig 1A shows the adopted arrangement of the vegetation types, based on a schematic representation of the actual river systems in the study area. It was taken that small rivers must traverse seven cells before becoming a medium river, which must then traverse 21 more cells before becoming a large river. Interfluves were typically two cells wide, but wider on the main watershed.

### Natural carrying capacities and death rates

The habitat associated with the large rivers was regarded as the most favourable, having a standard, natural, carrying capacity (SNCC) per km$^2$ of 5000 adult females and 2500 adult males if there was no net invasion or emigration of flies. Given that the habitat forms only 10% of the

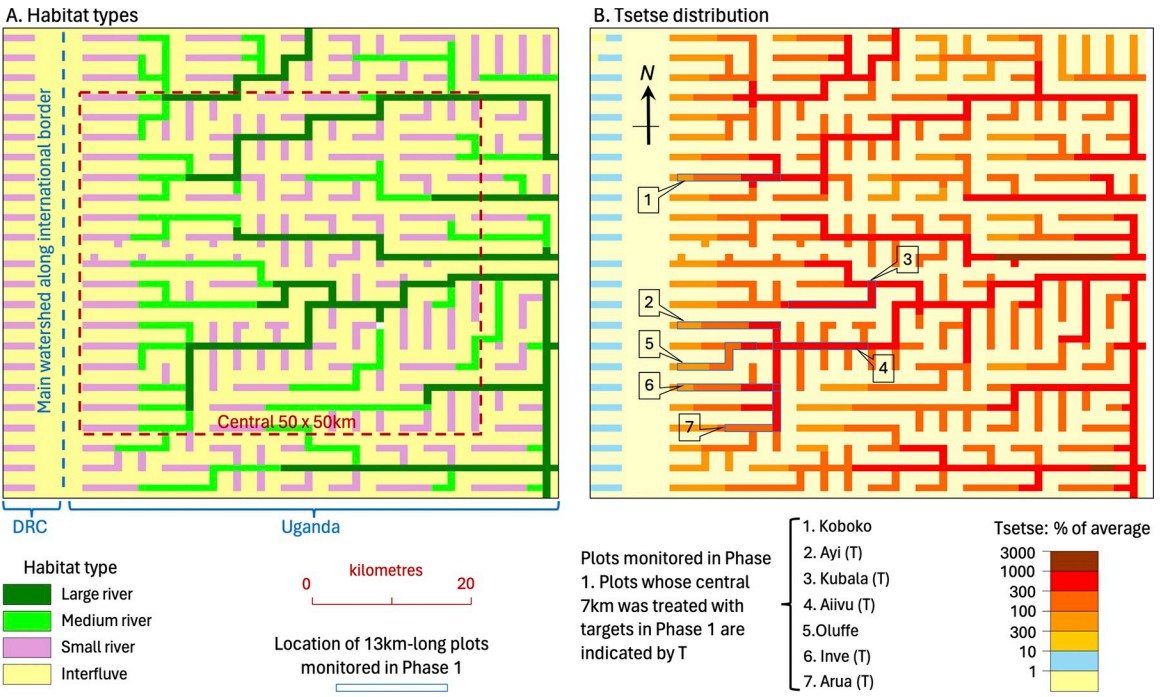

**Fig 1.** A: arrangement of the four habitat types in the model's simulation of the study area in NW Uganda. B: modelled distribution of male plus female tsetse resulting from the standard set of parameter values. Tsetse abundance is shown as the number per cell expressed as a percent of the average abundance in all 4900 cells of the whole map. That average is 49.1.

area of cells with a large river, the standard population in a cell with a large river was 500 females and 250 males. The adopted set of parameter values needed to stabilise the population at that level is shown in S1 Table. These parameters were based as far as possible on published data, with the compatible level of natural adult mortality being found by Excel's Goal Seek facility. The habitats of other rivers and interfluves were made less favourable by increasing the death rates of adults and pupae. Unless stated otherwise, each of these death rates at SNCC was increased by factors of 1.05, 1.10 and 1.20 for the medium rivers, small rivers and interfluves, respectively. This meant that in the absence of control and any net movement, and allowing for habitat cover, the natural carrying capacities for cells with a medium river, small river, or interfluve were 92, 13 and 0 males, and 184, 25 and 0 females per cell, respectively. However, during the simulations the numbers of tsetse in cells with relatively poor habitat could be increased a little if those cells were next to cells with better habitat and hence relatively many flies, which formed an invasion source. Hence, in heterogeneous terrain there were never zero flies in the interfluve cells.

All of the above population numbers are arbitrary, since no data are available for the true density of tsetse in NW Uganda. However, while the absolute numbers adopted by the simulations are suspect, it is pertinent to focus on their relative values. This means, for example, that the location of the concentration points of the tsetse population, and the percents of population control, are unaffected.

The natural death rates of adults and pupae in any habitat type were made density dependent, based on the formula:

$$m_d = m_s((1-k) + kd/s)$$

where:

$s$ is the density at the SNCC

$d$ is the current density per km$^2$

$m_s$ is the natural death rate for any given sex and age class in the given habitat type, when the density in that habitat is at the SNCC;

$m_d$ is the natural death rate for any given sex and age class in the given habitat type, when the density in that habitat is $d$

$k$ is an input constant.

Unless stated otherwise, k was set at 0.1, meaning that when the current density declined from SNCC to zero, the death rate declined by a factor of 0.9. Density was always expressed as the total males plus females per km$^2$ when dealing with the death rates of either male or female adults, and as the total pupae of both sexes per km$^2$ when dealing with pupal mortality. For convenience, the death rates of immature stages, comprising eggs and larvae, were taken as zero in all habitats at all fly densities, since the actual death rates of these life stages seem <1% per reproductive cycle of 10 days [14].

## Fly mobility

Daily diffusive movement was simulated by transferring flies between orthogonally adjacent cells, it being considered that before and after the transfer all flies occurred at the centres of the cells they occupied. Unfortunately, while this sort of orthogonal movement is convenient and manageable to model, it is imperfect in the following two respects. First, it was impossible for the modelled flies to transfer to a diagonally adjacent cell in one day, although they could get there after two days of movement. However, that would mean that to get to a diagonally adjacent riverine cell the flies would have to go through an interfluve cell. Thus, to ensure that flies in a riverine cell could transfer directly to adjacent riverine cells in a single day, all riverine cells had to be connected to each other orthogonally in the model's map, implying that all rivers had to flow either parallel or at right-angles to each other (Fig 1A). Second, since we use cells of side 1km and a pattern of daily orthogonal movement which involves flies travelling only between the centres of orthogonally adjacent cells, the error in simulating various rates of diffusive movement over many days can be great if cell evacuation rates are set to give the right average movement over just a few days. We minimised this problem by setting the rates of cell evacuation such that they produced the required average rate of daily diffusion assessed over 30 days.

In addition, there was the problem that the markedly heterogeneous vegetation in NW Uganda is believed to affect fly mobility. In allowing for this, it was imagined that when daily movement began, all flies in the cell, i.e., at its centre, started to move equally in all four orthogonal directions. Of the flies travelling in each direction, some reached the edge of the cell and transferred to the centre of the orthogonally adjacent cell there, whereas all other flies returned to the centre of the cell in which they started the day. It was taken that for the transfer of flies from cells with relatively extensive habitat to riverine cells with less habitat, the cell evacuation rate was reduced in proportion to the reduction in habitat abundance. For example, if the flies were moving from a cell containing a large river, with 10% habitat cover, to a medium river, with 7% cover, the normal evacuation rate from the large river was reduced by a factor of 7/10 = 0.7. If the flies were about to transfer from a cell with a river to an interfluve cell, then unless stated otherwise the evacuation rate was reduced by a factor of 0.01, implying that 99% of the flies that would have left to enter the interfluve did not do so, but turned back to stay near the river. Flies on the point of transferring in the reverse direction, i.e., from an interfluve

to a cell with a river, or from a small to a larger river, never showed any reduction in evacuation rates, being assumed always ready to enter any relatively favourable habitat they encountered.

The computations of all the varied rates of cell evacuation involved in the above principles depended on first deciding on the daily rate of two-dimensional diffusive movement in a landscape conceived as composed entirely of the best habitat, i.e., with all cells covered completely by the sort of habitat normally found near the banks of a large river. Unless stated otherwise, it was taken that recently emerged females with poorly developed wing muscles [15] moved an average of 200m/day, increasing linearly to a maximum of 400m/day after 10 days as the muscles developed. Thereafter, the mobility remained steady. For males, which are known to be less mobile [16], the mobility in each age class was halved. These levels of mobility are compatible with the field indications for the overall rate of dispersal of *G. f. fuscipes* in a large block of mostly favourable habitat in SE Uganda [7].

## Control by targets

Unless stated otherwise, Tiny Targets were deployed only in cells that contained a river, and always at the rate of 20 per cell, corresponding roughly to the standard deployment rate adopted in Uganda, i.e., 20 per km of riverbank [5, 9]. Adopting the standard degree of habitat cover per cell, specified above, the number of targets per $km^2$ within the habitat of the cells with large, medium and small rivers was 200, 286, and 500, respectively. For special studies in which the habitat cover within the cells was varied away from the standard, the density of the targets was considered to change accordingly.

It was taken that Tiny Targets in good order, recently deployed at a density of one per $km^2$, would produce a daily kill rate of 0.01% of the population of females of age one day. The rate rose linearly to be five times greater, i.e., to 0.05%, for females ten days old, and then remained at that rate for all older females. This rate was chosen since one large, odour-baited target kills around 2% of the population of *G. pallidipes* per day [17]. The greater size of such a target, combined with its odours, might be expected to make it around 40 times as effective as a Tiny Target for *G. f. fuscipes* [18–20]. It was taken that rates for males of any age were 80% of those for females of the same age. This schedule of mortalities for the sexes and age classes was termed "Schedule 0.05", being named in accord with its kill rate of 0.05% for female flies of age ≥10 days. Unless stated otherwise, Schedule 0.05 was adopted as the standard in all modelling. Other schedules were sometimes employed, each of which involved increasing or decreasing the rates for each sex and age by a set proportion, and each schedule was named according to its rate for the old females. For example, Schedule 0.01 involved the kill rate of 0.01% for such females.

Studies in Uganda indicated that many targets became damaged, stolen, or removed by floods, so that only around 20% remained after six months [5]. Moreover, it is expected that much of the insecticide deposited on the targets was weathered away in that time [9]. The combined effect of all of these processes was modelled as the "degradation rate", covering a first order decline in the killing power of targets. Unless stated otherwise, the degradation was at the standard rate of 1.5% per day, which meant that the efficacy declined to only 6% of the initial level after six months. After that time, i.e., at the beginning of the 184[th] day after previous deployments, any of the old targets remaining were regarded as removed and a full set of 20 new targets was deployed per treated cell. Such six-month refreshments occurred regularly for as long as the target operation continued, in keeping with practices in Uganda [5, 9]. Given that the target degradation in field situations was very variable [5], simulations were performed with a range of degradation rates.

## Simulation procedure

For simulation of control, Excel was put into iterative mode with each iteration being taken as covering one day. The calculations of each iteration started at the top of the spreadsheet and travelled in a wave down the sheet and through the lifetables. The first action in the wave was to advance the date by one day, followed by moving the flies up one compartment of the tables, that is to a compartment for flies a day older. During that transfer the flies were subject to natural mortality and movement, and then to the control mortality due to targets. Finally, female tsetse dropped the day's crop of larvae, which promptly pupated, and pupae that had completed their development produced new adults.

The first set of simulations was aimed at getting appropriate values for those input parameters that are the most difficult to quantify from published sources. Such work began by considering the control produced by a range of mortality rates imposed by Tiny Targets that degraded at various rates and were operated in situations of homogeneous habitat with no net movement of the flies. It continued by testing the interplay of various levels of habitat cover, density dependence, fly movement, and target degradation in governing the distribution of tsetse in the schematic map of the heterogenous landscape in NW Uganda. For this the focus was primarily on the extent to which the simulations could reproduce the field effects of the one year of control operations performed in the small plots studied during Phase 1 of the work of Hope et al. [5] (Fig 1B). It was appropriate to give special attention to those plots since they are the places subject to the most detailed study in the field [5, 9]. Having produced seemingly credible simulations of the control achieved in Phase 1, further simulations were performed in succession to assess how well the model could represent the effects of the larger scale control performed during the succession of two-year periods forming Phases 2–5 of Hope et al. [5].

Before simulating the control, it was necessary to produce a map of the stable uncontrolled distribution of tsetse. Such maps were always created by seeding flies into all cells of the map and then running 10 000 iterations with no control, so that the numbers of flies in each cell adjusted to the combined effects of the mobility and natural mortality of tsetse, and habitat abundance, throughout the map. Fig 1B shows the distribution of the uncontrolled population arising from the standard set of input parameters. The age structure of the stable population is shown in S1 Fig.

## Outputs

The output of prime interest was the change in the simulated number of flies in the operational areas, because this can be expected to correlate with changes in trap catches, i.e., the main type of data available for validation from the field work and the main statistic by which the degree of tsetse control is usually judged [5,6,9,21]. For simulations of tsetse abundance in representations of the actual operational area in NW Uganda, the numbers of male and female tsetse were usually pooled, and the extent of control was averaged over the six months (183 days) following the last refreshment of the targets. This type of statistic accords with the form of the empirical results reported for operations using Tiny Targets in NW Uganda [5]. In other simulations the male and female numbers were separated and often reported daily.

Records were also made of changes in the simulated age structure of the population since at some times and places the field work studied the ovarian age categories of females in the trap catches. Not surprisingly, control by targets reduces life expectancy and so decreases the proportion of older flies in the population [13,22]. The complication here is that fly age affects the availability to stationary traps and targets, presumably because young flies are relatively immobile and so tend not to encounter such baits [15]. As discussed above, it was assumed that the poorly mobile flies were <10 days old. Hence, to assist the interpretation of what overall trap

catches mean about the density of the population, and to assess the overall kill rates imposed by targets, it was necessary to record the simulated daily percentage of the population that was aged ≥10 days. However, that statistic was of little use when validating the field data for the age structure of trap catches since virtually all the catches were in that age group. Hence, the modelling also simulated the percent of the population that was >40 days old. That age is of particular interest since the ovarian method used to age the field caught females in Uganda could not identify precisely the age of flies older than about that [23]. Lastly, given that a reduction in the proportion of older flies will decrease the percentage of flies of breeding age, modelling recorded the rate of pupal production.

## Results

### Parameterisation

**Effects of kill rates with targets always in good order.** To expose the basic abilities of targets to control tsetse in the absence of invasion, the model ran targets for a year in a landscape of effectively infinite extent, consisting only of cells with large rivers, and with no target degradation. The population data were produced for a cell at the centre of the map, i.e., a place where there could be no net invasion to offset the flies killed by the targets. These simulations showed that an increase in the kill rates, from Schedule 0.01 through to Schedule 0.10, enhanced greatly the rate of population reduction in both sexes (Fig 2A and 2B). The rough rule of thumb is that, in the absence of invasion, the time taken to achieve any given level of control is inversely proportional to the kill rate. However, the degree of population decline with time was unsteady in the early stages, especially when the kill rate was high, because flies continued to emerge from pupae in a constant stream until a pupal period had been completed. That phenomenon had lasting repercussions, involving continued fluctuations in the percent of old flies in the population for several pupal periods, especially when kill rates were highest, i.e., with Schedule 0.10 (Fig 2C and 2D). However, such great fluctuations were largely theoretical since they occurred within a population that was so sparse as to be virtually extinct. The degree of population decline associated with Schedule 0.05 seemed the most applicable to the field situation in NW Uganda, since it produced the maximum rate of decline evident there soon after target deployment, i.e., when target degradation and invasion were minimal [5]. While the selection of the appropriate kill schedule was the prime concern in the parameterisation of target efficacy, the changing age structure of the population caused two other points of interest to emerge, as below.

First, the cyclic reductions in the proportion of flies ≥10 days old produced synchronised decreases in the percent of the total population killed per day (Fig 2E and 2F), because such old flies were the most available to targets. Consequently, the average percent of the total population killed per day, once control had started, was always less than the percentage on the first day of target deployment. For example, with Schedule 0.01 the kill rate for females on the first day of control, and the average daily kill rate for the rest of the year, was 0.0091% and 0.0089% respectively. The corresponding figures were 0.0456% and 0.0393% for Schedule 0.05, and 0.0912% and 0.0692% for Schedule 0.10. That is, the difference between the kill rates on Day 1 and subsequent days was small when the kill rate on Day 1 was low, but the difference increased as the Day 1 rate rose. Looked at another way, the general principle is that the difference between the kill rates on the first and subsequent days is negligible for campaigns in which the population declines to around 1% in a year, but starts to be much greater in operations involving population declines to about 0.0001% in a year. Allowing that this principle depends on the timings of the reproductive cycles, which are roughly the same for all tsetse species, it can be taken that the principle will apply to any of the species.

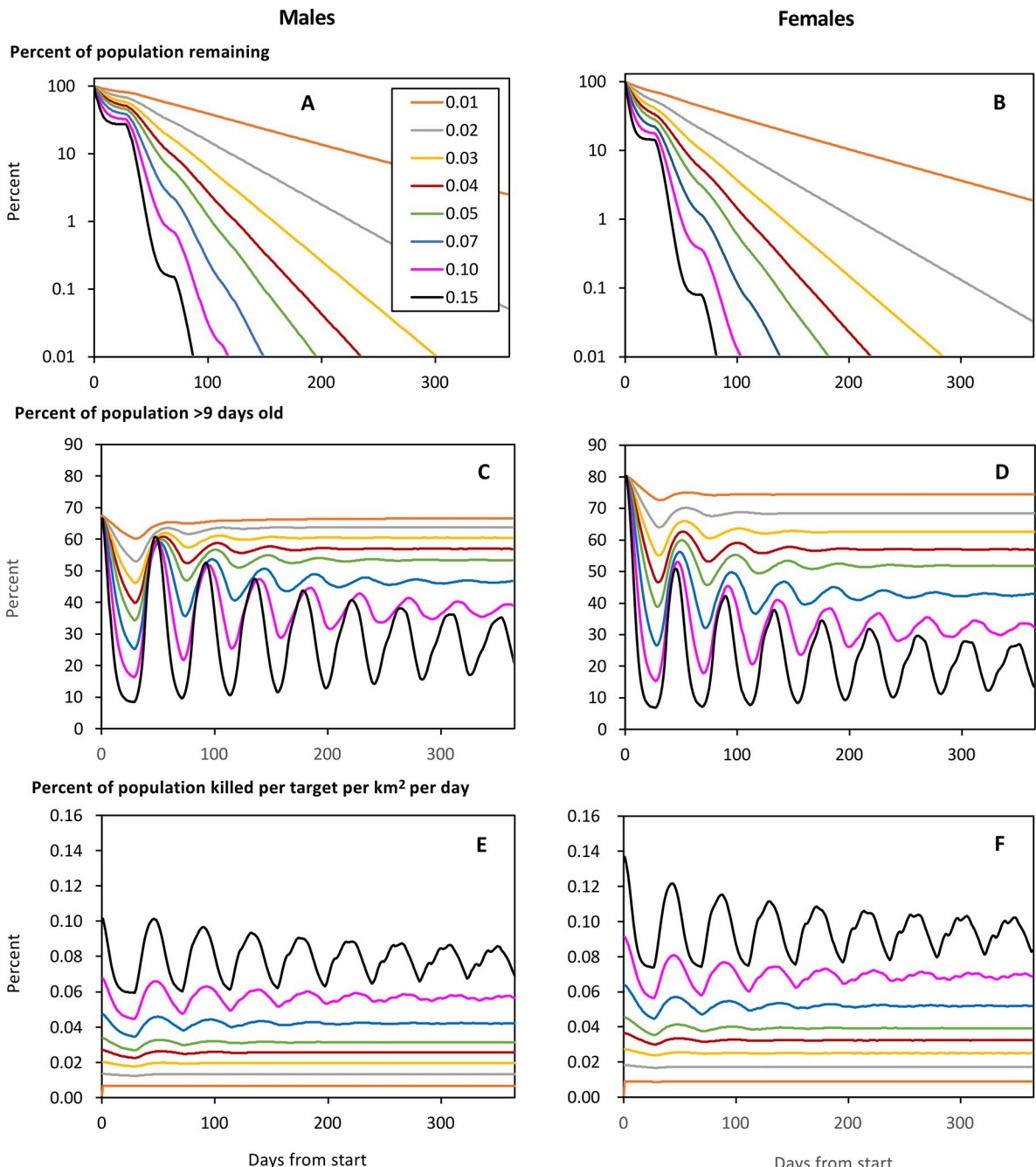

**Fig 2.** Modelled percent of the initial numbers of male (A) and female (B) tsetse remaining after various days from the start of target operations associated with the various kill rates of Schedules 0.01 to 0.15, characterised by the daily kills of 0.01% to 0.15% per target per day among females ≥10 days old. Also shown is the associated percent of the male (C) and female (D) population that is ≥10 days old, and the percent of the male (E) and female (F) population killed per day, for all age classes combined. The targets were subject to no degradation and were deployed for one year in a model landscape consisting only of the best habitat, i.e., cells with a large river.

Second, the breeding rate declined during control because only the older flies produce pupae and such flies become relatively scarce. For example, on the day before control began, the daily number of pupae produced per 1000 females was 77.8, but it dropped to an average of 71.3 in the year of control associated with the modest kill rates of Schedule 0.01. The

corresponding figures for the greater kill rates of Schedule 0.05 and Schedule 0.10 were much lower, being 46.6 and 26.0, respectively. The pertinent principle is that the efficacy of control by targets is not due simply to an increase in death rates, but also to a potentially drastic decline in birth rates, albeit that the change in pupal production has no instant impact on trap catches of adults.

**Effects of target degradation.**   Using the map employed in previous work, consisting entirely of cells spanned by large rivers, targets were employed with the kill rates of Schedule 0.05 on the first day, and with rates of target degradation ranging from 0.0 to 2.0% per day. As expected, the decline in tsetse numbers was relatively low for relatively high rates of degradation (Fig 3A and 3B). The results for the 1.5% degradation rate are most pertinent since that is roughly the rate at which the targets were observed to degrade in operations conducted in Uganda [5]. Moreover, modelling with that degradation rate simulated levels of population decline occurring at the time when degradation rates were assessed. The population decline involved was down to an average of around one percent of flies remaining during the second period of target redeployment, 6–12 months after control first started [5].

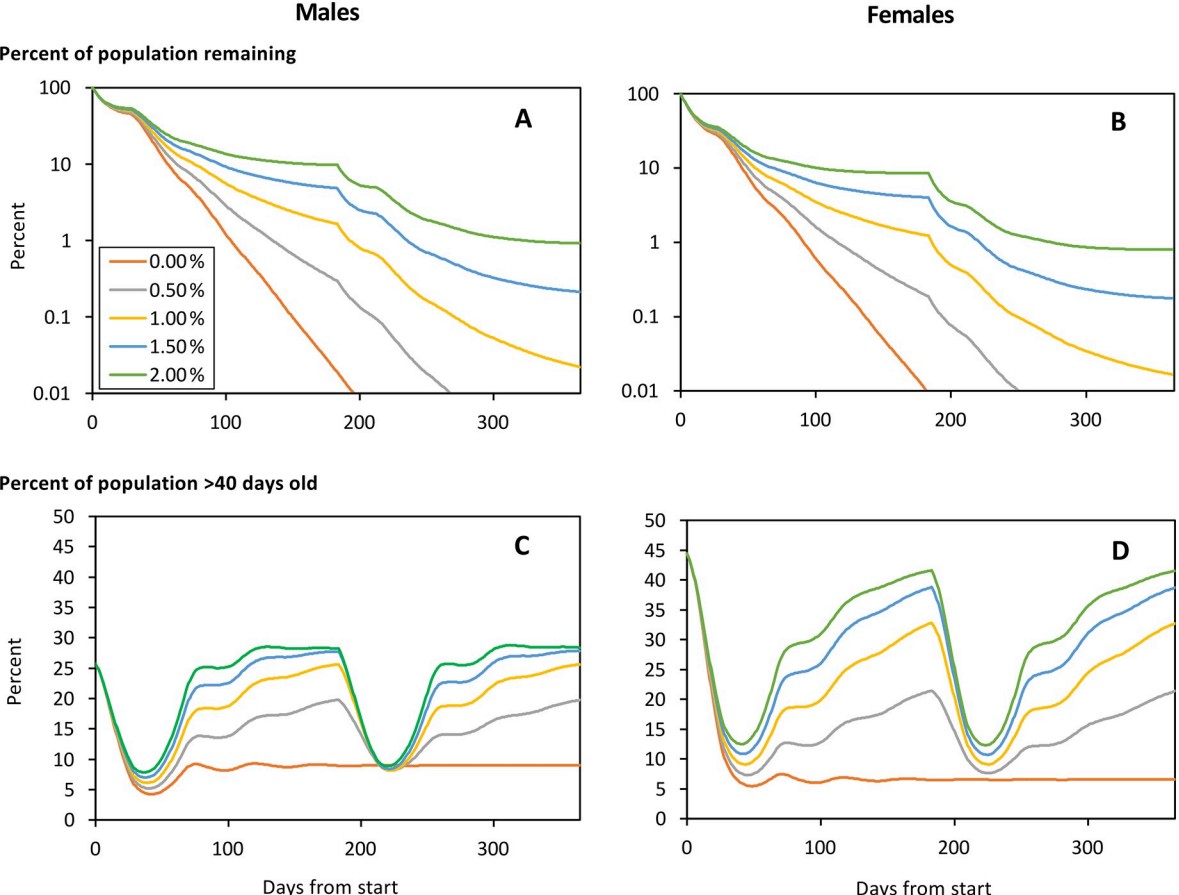

**Fig 3.** Modelled percent of the initial numbers of male (A) and female (B) tsetse remaiming after various days from the start of target operations associated with rates of target degradation of 0.00% to 2.00% per day. Also shown is the percent of the male (C) and female (D) population that is >40 days old. On the first day of control the targets produced the kill rates of Schedule 0.05, associated with a kill per target per day of 0.05% among females ≥10 days old. The target campaign ran for one year in a model landscape consisting only of the best habitat, i.e., cells associated with a large river. Target deployments were refreshed after the first six months.

Not surprisingly, the percent of the population that was old declined sharply at the start of the target operation and immediately after the refreshment at six months. At those times there was little effect of the different rates of degradation since the degradation had by then had little time to become apparent. If there was no degradation of targets with time, the percent of old flies remained very low throughout the rest of the simulation period, there being only slight undulations while the relationship between births and deaths stabilised. For rates of degradation >0%, the percent of old flies began to rise after about 50 days when the degradation was well under way, allowing the tsetse population to start increasing. With degradation rates of around 1.5% the percent of old flies was high for much of the six months after refreshment. Thus, even in the present subset of simulations, which were organized so that the percent of old flies could not be enhanced by invasion, the indications are that it could be difficult to notice a marked change in age structure averaged over many months.

**Exploring parameter values associated with Phase 1 of control.** All the simulations for this work employed the vegetation map intended to represent the various river systems and interfluves in NW Uganda, with the targets deployed for one year in the five separate 7km-lengths of river treated in Phase 1 of Hope et al. [5] (Fig 1B). In accord with the field work, the tsetse population was monitored in the cells of each treated length, together with the groups of three cells representing the 3km of river upstream and downstream of each treated length. This meant that the monitoring was along 13km-lengths of river.

Data produced using standard input values showed, in keeping with field observations [5], that the tsetse population was reduced by around 99% in the centre of the treated lengths of river (Fig 4). The degree of control decreased towards the ends of the treated lengths, but control was evident in the 3km beyond the ends. Also, in keeping with field observations [5], the degree of control upstream was greater than downstream, i.e., the average percent of the tsetse population remaining in the six cells upstream of the central cell was 25.9%, as against 36.0%

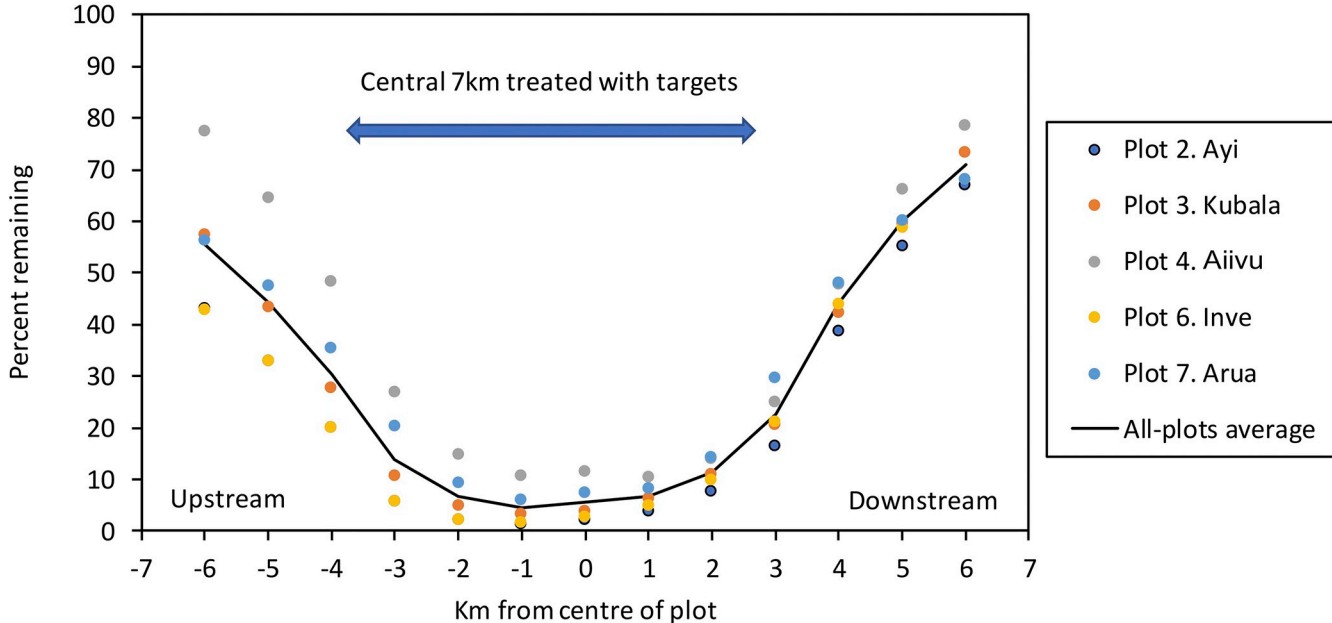

**Fig 4. Simulated percent of the initial male plus female population remaining along each kilometre of the five 13-km lengths of river associated with target treatments in Phase 1 of the field work in NW Uganda.** All parameters were at the standard values. The target campaign ran for one year with the targets being refreshed after six months. The percent remaining in each plot was assessed as the average of the daily percents in the last six months. Data for Ayi and Inve were so similar that the yellow plots for Inve totally obscure some of the plots for Ayi.

in the six downstream cells. Credible variations to the values of the parameters used in the simulations showed, as expected, that changes which directly or indirectly affected the natural death rates altered the natural abundance of tsetse by up to three-fold, but none of the changes produced any great alteration to the upstream/downstream distinction in the degree of control (S2 Table).

## Simulations of population reduction in Phases 1 to 5

Fig 5 shows the target treated rivers and the simulated average percent of the initial tsetse population remaining in the last six months of each of the five phases of operations in NW Uganda. The data for each phase are considered in turn, below.

**Phase 1.**   Although monitoring was performed along all 13km of the seven riverine plots studied in Phase 1 (Fig 1B), targets were deployed only in the central 7km of just five of these plots (Fig 5, Phase 1). In accord with the field results, the remaining population was low along and near the riverine areas treated by targets. However, there was relatively little impact on the tsetse population in those untreated riverine areas, i.e., in the Koboko and Oluffe plots, that were not connected directly and closely to the target-treated rivers. The detailed outputs show that at Koboko, many kilometres away from any targets and on a separate river, the mean percent remaining along the monitored 13km was 100.0%, indicating no control. At Oluffe, which is only 3km away from treated rivers to the North and South, the percent remaining was reduced a little, to 98.2% (range 92.3–99.7) with the degree of reduction being greater on going downstream within the 13km, i.e., towards the nearby connected parts of rivers that had targets. Such small degrees of control are attributable to the tsetse diffusing away from the Oluffe plot and not being fully balanced by the potential invaders that had been killed in the target-treated areas several kilometres distant. Not surprisingly, and in keeping with the deliberate programming of the model, tsetse diffusing away from the Oluffe plot tended to move mostly down the river, rather than across the interfluve.

Although the simulations of Phase 1 indicated that movement had little effect on catches when the immediate route of invasion was through interfluves, the situation was very different when the flies could move directly along rivers. For example, in the simulated Phase 1, there were seven target-treated cells covering the large river at Aiivu (Fig 1 B, Plot 4). Each of these cells was connected to another riverine cell on at least two sides, so allowing strong movement. In the last six months of the phase the simulated percent of flies remaining in the treated cells averaged 16.2%, with a range of 10.3–26.9% among the individual cells. This average is 24 times greater than the 0.7% remaining in each of the seven cells when the same treatment system was used for a simulation in which no movement was allowed.

**Phase 2.**   This phase involved expanding the target deployments a little, mostly towards the North and East, and was associated with a general improvement in the level of control. The Oluffe plot, which was untreated in Phase 1, and which registered poor control then, was treated in Phase 2 and showed a marked increase in the rate of control. Far to the North, the Koboko plot continued to show no control. The results in each plot accord with field data [5].

**Phase 3.**   For this phase the target deployments were expanded much further North and East, so producing the most extensive deployments of any phase. The previously untreated river at Koboko in the far North was included in these deployments and, in keeping with field results, the population there declined markedly.

**Phase 4.**   The overall extent of the target deployments was slightly reduced in this phase, due mainly to fewer placements in the North. However, this was partially offset by minor increases in deployments in the South-East and far West. The general level of control was not affected greatly by these changes in deployment, consistent with field results [5].

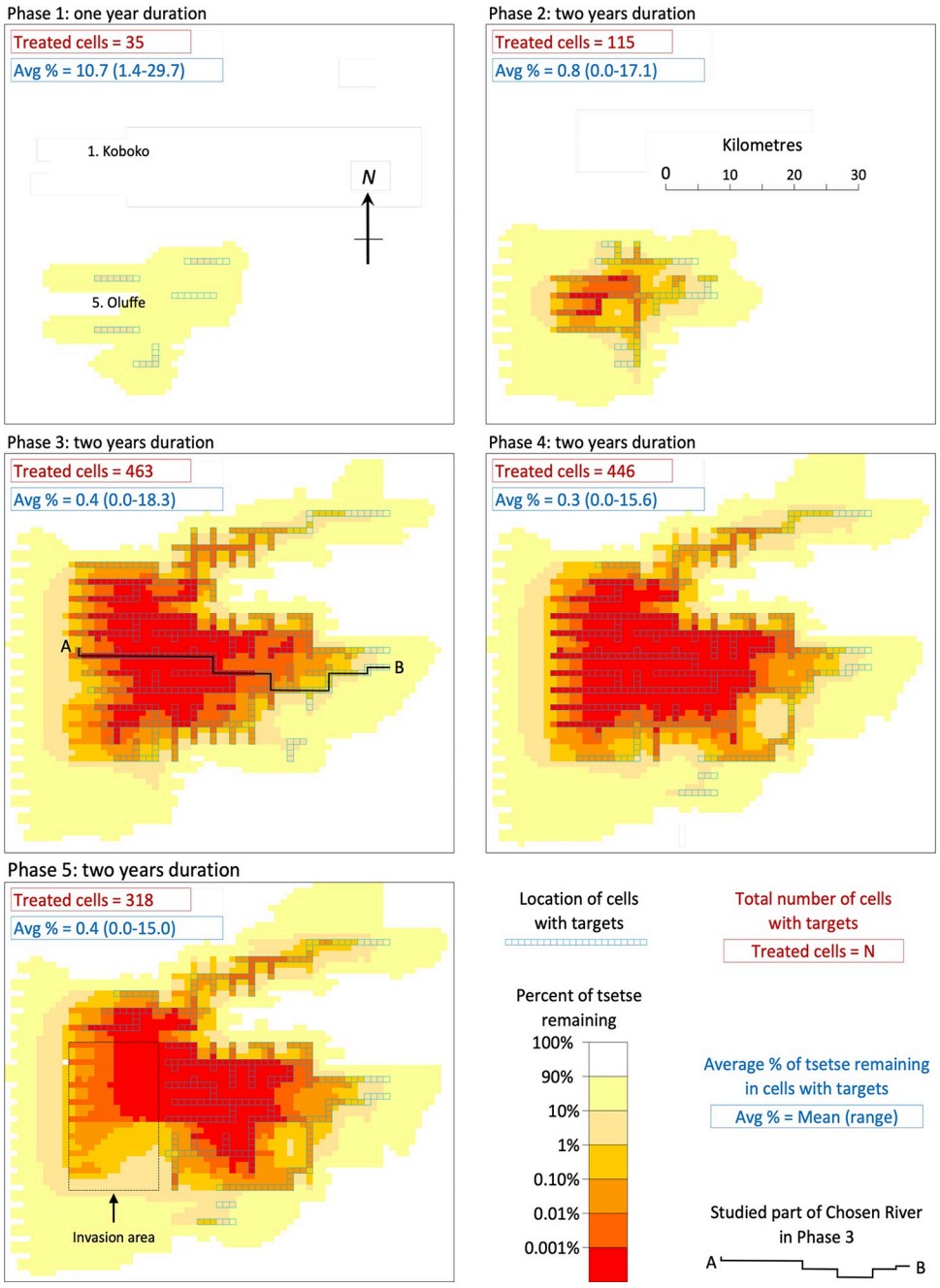

**Fig 5. Distribution of targets and the average percent of the pre-control population of tsetse remaining during the last six months of control, in the simulations of the five phases of control associated with the field work of Hope et al.** in NW Uganda [5]. Also shown are the numbers of cells with targets in the simulations, and the outputs for the average percent of tsetse remaining in the cells with targets. Areas identified in black font are mentioned in the text.

**Phase 5.** Anticipating a phased scaling back of tsetse control, we simulated the effect of halting the deployment of Tiny Targets in the West and South-West, in an area designated the invasion area (Fig 5, Phase 5). The simulations indicated no marked re-bound in the tsetse population in that area: the abundance of tsetse remained low, especially in the North-East of the invasion area where the target treatments were close on two of its sides.

### Spatial details of the abundance and age of flies in Phase 3

Because Phase 3 involved the greatest treated area, it is pertinent to consider in detail the spatial aspects of the control it produced. However, the simple expedient of examining the population remaining along a straight transect through the treated area is inappropriate because such a transect would pass through many interfluve cells in which the population was very sparse even before any control. Hence, it is most pertinent to consider the population remaining along a single river, even if the river meanders a little from a straight transect through the operational area. The river selected, called the Chosen River, was that which has its source at Point A of Fig 5 (Phase 5) and flows out of the treated area at Point B, before continuing in a roughly straight line to the eastern edge of the map.

**Effects on degree of control.** The salient outcome of the simulations along the Chosen River was the fact that the degree of control there was very high using the standard kill rates of Schedule 0.05 and the standard degradation rate of 1.5% per day. Thus, after completing the five years of Phases 1, 2 and 3 operated in sequence, virtually no flies remained in the middle sections of the treated part of the river, representing a 99.99999% control there (Fig 6A, Schedule 0.05 and Fig 6B, 1.5% degradation). In contrast, the actual levels of control achieved in the field campaign were of the order of only 99.0% [5]. Hence, while kill rates of Schedule 0.05 combined with the degradation rate of 1.5% had seemed to produce good simulations of the field results of Phase 1, such rates gave control that was up to five orders of magnitude too great in Phase 3. The implication seemed to be that by the time that Phase 3 was reached in the field the kill rates could have declined and/or the degradation rate could have increased. To assess how much the rates might have altered, simulations were run in which the standard Schedule 0.05 and the 1.5% degradation were applied for the combination of Phases 1 and 2, followed by the changed rates for the whole of Phase 3. These changes involved keeping the degradation rate at the standard 1.5% and varying the kill schedule, or keeping to the standard kill schedule and varying the degradation. The results showed that to simulate the average field level of control of around 99%, the kill rates had to be reduced to about a third or the degradation rates had to be enhanced about three-fold (Fig 6A and 6B). Adopting the kill and degradation rates that gave realistic degrees of overall control, the percent control near the source of the river was around one order of magnitude greater than at the extreme downstream end of the treated section.

To emphasise that there appeared to be a substantial change in the dynamics of control between Phases 1 and 3, simulations of Phase 1 were rerun with the low kill rates or high degradation rates that were needed to account for the relatively low levels of field control at the end of Phase 3. Such simulations showed that the average percents of males plus females remaining in the target treated areas of Phase 1 were 53% and 35% for the kill rates of Schedules 0.01 and 0.02, respectively, and 38% and 43% for the degradation rates of 4.5% and 5.5%, respectively (S3 Table). Thus, while low rates of kill and/or high rates of degradation were needed to simulate the field results of Phase 3, such rates adopted for simulations of Phase 1 produced degrees of control that were one to two orders of magnitude poorer than the field results of Phase 1 [5]. The implication is that the basic efficacy of the target operation in Phase 1 was indeed substantially better than in Phase 3.

**Age and abundance of females.** Simulations along the Chosen River gave outputs for the number of females per km$^2$ of habitat (Fig 6C and 6D) and for the percent of the female population that was >40 days old (Fig 6E and 6F). The results indicated the theoretical principle that, in operations of the Ugandan type, the percent of old females could decline by around half, but only in places where the numbers of females per km$^2$ had reached such low levels that the likelihood of catching a fly for age studies would be virtually zero. The results also illustrate

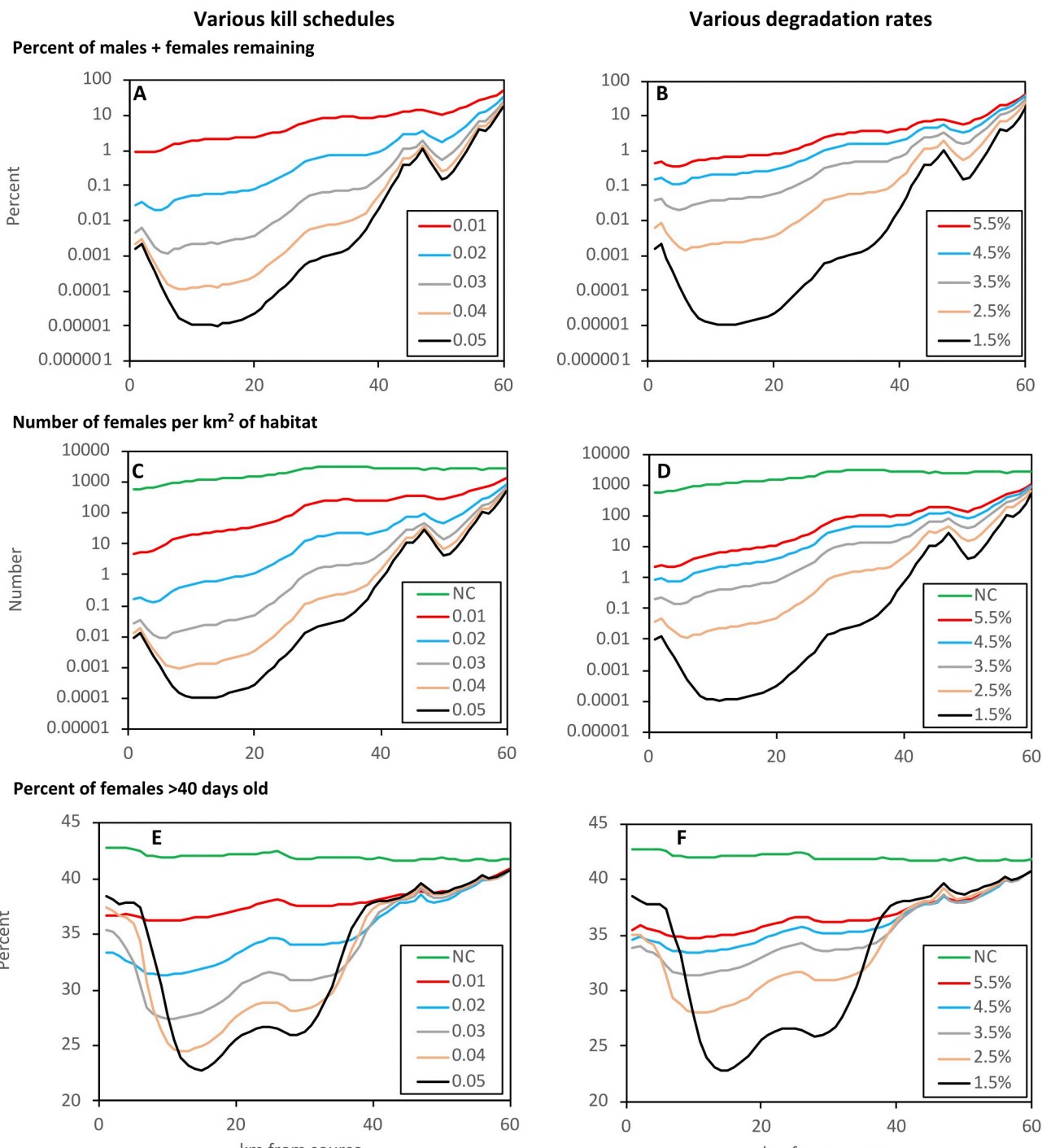

**Fig 6. Data for the tsetse population along the Chosen River for the last six months of control in the simulations of Phase 3 of the field work of Hope et al. in NW Uganda** [5]**.** The data cover the average percent of males plus females remaining (A,B), the average numbers of females per km² of habitat (C,D) and the average percent of females of age >40 days (EF). For A, C and E the degradation rate in the whole of Phase 3 was held at 1.5% while the target treatments involved kill shedules of 0.01 to 0.05. For B, C and F the kill shedule was fixed at 0.05 while the degradation rate varied from 1.5% to 5.5%. Each of the variants of control implemented in Phase 3 was preceded by the standard control in Phases 1 and 2, comprising the kill shedule of 0.05 and 1.5% degradation. The graphs identified as "NC" refer to data before any control, i.e., prior to Phase 1. The plots start at the source of the river, near the main watershed, and go downstream in a predominantly eastward direction. Irregularities in the graphs correspond to points at which the size of the river changes, or tributaries enter. Notice that the scale of the Y axis on E and F starts at 20%, not zero.

the principle that the percent of old females tends to be highest near the ends of the treated part of the river. This is because the population at the ends is maintained largely by invasion, and invaders tend to be old.

In general, the simulations showed that where the remaining females occurred at densities of $\geq 1/$ km$^2$ of habitat, i.e., where there was at least a moderate chance of catching females for age studies, the percent of females that were $>40$ days old was high. For example, using the standard set of parameters to simulate Phase 3, thus giving the distribution of flies shown in Fig 5 (Phase 3), only 83 of the target-treated cells satisfied the minimum criterion for female density. For these cells the mean percent of old females in the remaining population was 40% (range 36–44%). When the number of target-treated cells satisfying the minimum density requirement was increased to 443, by using the low kill rates of Schedule 0.01, the mean percent of old females was 38% (36–44%). All of these percents differ little from the average of 43% (41–44%) on the whole map before control. Such small differences would be difficult to detect with confidence in the field, consistent with the fact that the field work in NW Uganda gave no evidence that the targets changed the age structure of the tsetse population [5].

## Discussion

We developed a spatially explicit model of the dynamics of a population of tsetse to answer questions about three empirical aspects of deploying Tiny Targets to control *G. f. fuscipes* in NW Uganda.

1. Why was the impact of Tiny Targets greater in the upstream sections of rivers and streams?

The model suggested that several factors were responsible for this: (i) the invasion sources near the watershed were less well populated than those downstream, (ii) natural death rates were relatively high upstream, so that the population there had little density-dependent resilience, and (iii) suitable tsetse habitat was less abundant upstream, so that the density of targets per km$^2$ of habitat was relatively high, thus imposing a collectively greater death rate on the population.

2. While the targets rapidly reduced the abundance of tsetse to low levels, why did they not eliminate the tsetse population?

Modelling suggested that failure to achieve local elimination of the tsetse population is not a consequence of targets being lost or degraded but rather the invasion of tsetse from neighbouring untreated areas.

3. Why did the targets not produce a noticeable reduction in the mean age of the population.

Modelling indicated that the direct effect of the targets on reducing the abundance of old flies was evident mostly in the few weeks following the target deployment, after which the old flies could be replenished during several months of invasion and re-instated longevity.

In anticipation of the scaling back of tsetse control operations, we also used the model to assess the rate and extent to which tsetse populations would recover following the cessation of vector control. The model predicts that a limited scale back of control operations in the headwaters of a drainage system will not lead to a marked and rapid re-bound in the tsetse population: the abundance of tsetse remained low, especially where neighbouring areas are still subject to vector control.

We used the model to answer questions arising from operations conducted to control *G. f. fuscipes* in NW Uganda. Given the rough correspondence between the simulations and field experience, it seems that the model can be an aid to predicting the effects of various sorts of

target campaigns and teaching the principles on which such campaigns depend. It is potentially important, therefore, that the model predicts that the tsetse population near the headwaters will not rapidly rebound to its initial level when the target operations are scaled back there. This is partly because the modelling, and field experience, show that the degree of control there tends to be high. It is also because the modelling allows that the habitat near the headwaters is relatively sparse and unfavourable.

## Caveats

The model is simplistic in that it is non-seasonal, portrays the vegetation schematically, simulates diffusive movement in an orthogonal manner, envisages that target deployments take place all at once on a single day, and assumes that subsequent degradations of the targets occur evenly in time and space. Moreover, while the model's parameters for the timing of reproductive events, and for the general levels of mobility and natural death, must be roughly right [7, 9], the detail for the effects of vegetation on the death rates and movement are largely arbitrary. Given these limitations, there is need for caution in several respects, as below.

First, riverine tsetse in different operational areas are liable to have distinctive variations in their seasonal dynamics near the watersheds, especially since the rivers there tend to become dry at certain times of year. The importance of this is indicated by data from the Ugandan work at Koboko (S1 Text). Those data suggest that over the year the apparent density of tsetse near the watershed was on average about a quarter of that downstream. Moreover, the data also indicate that the seasonal variation in abundance became proportionally twice as great on nearing the watershed. Such phenomena imply that many of the flies upstream die off rapidly in the dry season, and /or that they migrate downstream then. This could explain why the field results showed that the control upstream of the target placements in Phase 1 was about 10-fold greater than downstream of the placements. Maybe the flies upstream moved down to the target-treated sections during the dry season and, being killed there, they could not return upstream in the wetter weather. In any event, it does seem that the failure of the present model to address seasonal issues could explain why the model's prediction for control near the watershed was less than observed at such places in the field. While the model is limited by the absence of seasonal variation in the abundance and age structure of tsetse populations, such variation is relatively limited for riverine tsetse in the equable climate of northern Uganda. For instance, the abundance of the tsetse population varied <10-fold (S1 Text). Savanna species can display much larger (>10-fold) variations in areas where there are marked seasonal changes in temperature and rainfall[24–26]. Modelling savanna populations would benefit from the incorporation of seasonality in model parameter estimates.

Second, although for the most part the standard set of input parameters simulated realistic degrees of control in Phase 1, the same set produced simulated control that was several orders of magnitude too great by the end of Phase 3. While it must be allowed that the flies could have evolved immunity to insecticide [27], there is no evidence that such resistance has affected target operations elsewhere. Moreover, if resistance had occurred by Phase 3 it would be expected that two years later, by the end of Phase 4, the resistance would have become particularly widespread, ensuring levels of control much worse that those actually observed in Phase 4. It seems more likely that the problem involved a reduction in the basic efficacy of the target campaign itself, perhaps associated with challenges in management and supervision when the scale of operations increased. In any event, the field experience and the simulations indicate that any such reduction in target efficacy need not necessarily prevent the achievement of those levels of control required to interrupt transmission. Moreover, in compensation for any drop in kill rates per target, or the greater loss of targets, it would help to reduce the normal field rate of

target degradation [5]. That would ensure that fewer targets would be needed or that control would be quicker, so easing the demands on management.

Third, the present arbitrary inputs for the efficacy of a single target, that is the kill rates of the various schedules, make sense only by assuming that the main occupiable habitat extends no more than about 20-50m either side of the river. This accords with the common experience of trap catches being very poor at greater distances from riverbanks or lake shores with narrow bands of lush vegetation [4]. However, in some situations the lush vegetation might be much broader, as near Lake Victoria in SE Uganda [7, 28]. Furthermore, the variation in habitat geometry might affect the availability to stationary baits [29] but, in any event, for places where the habitat is relatively broad it is likely that the number of targets per km of the riverbank or shoreline would have to be increased to maintain the abundance of targets per unit area of habitat.

## Theoretical considerations

The fact that *G. f. fuscipes* appears not to move much through the interfluves means that the population in the rivers is relatively isolated. This suggests that marked flies released by the river would be comparatively easy to recapture in operations limited primarily to upstream or downstream of the release points. Such mark-release-recapture (MRR) studies would provide valuable data for the death rates and mobility of tsetse in various parts of the river line. It would also allow estimates of population numbers. That would be useful in indicating the availability to the catching devices used for the MRR work. It would be especially informative in the present context if the catches from any killing device operated at the same time were related to the population density, thus helping the kill rates per bait to be established with greater confidence.

However, as present modelling shows, if kill rates per bait are evaluated like this they should be recognised as applying to the type and density of baits employed at the specific instant associated with the evaluation. This is because as control proceeds the age classes that are most readily killed by targets are the ones that decline most. However, the reduction in kill rate is no cause for practical concern, because it is the type of phenomenon that can become marked only when the kill rates are so high that the expected degree of decline in them is inconsequential. Moreover, while the change in age structure will reduce the kill rate, it will tend to compensate for that by also reducing the birth rate.

The known range of age-related issues, including the effect of age on mobility and natural deaths and births, makes it best to use models that allow for the age structure of the population Hence the present model, TP2, is an improvement on its early predecessor, Tsetse Plan 1 [13], which relied entirely on a rule of thumb relationship between the imposed death rate and population decline. TP2 is also better than another predecessor, Tsetse Muse [15], because it is fully two-dimensional. Moreover, TP2 allows consideration of a wider range of modelling than discussed here. For example, it can be applied to savanna habitats and different sorts of control measures used alone or in combination. It can also be used with inputs allowing for variations in local climate, different types of host, changes in the abundance and availability of hosts, and alterations to the behaviour of tsetse during the hunger cycle. Readers are invited to use and improve the model themselves.

## Conclusions

We developed a spatially-explicit model which simulated the movement and growth of a tsetse population and used this to identify the causes of three characteristics of vector control operations conducted against *G. f. fuscipes* in northern Uganda using Tiny Targets. The model's outputs lead us to conclude, first, that the impact of Tiny Targets varies according

to the topography of tsetse habitat; better control is achieved in places where habitat suitable for tsetse is limited. Second, the deployment of targets along only the larger rivers and streams and/or over a limited area allows invasion of tsetse from uncontrolled rivers into the controlled places. Such invasion prevents the local elimination of tsetse. While better control of tsetse might be achieved by deploying targets along all rivers and streams, the added cost may not be justified when elimination of transmission rather than tsetse is the aim of the intervention. Third, a limited scale-back of tsetse control operations does not lead inexorably to a rapid rebound in the tsetse population. If vector control continues in neighbouring parts of the drainage system, then rebound is likely to be limited. Putting together all these indications of the modelling, it seems that the model can indeed help with the planning and teaching of tsetse control.

## Supporting information

**S1 Fig. Numbers of tsetse per km$^2$ of habitat in various weeks of adult life, for a standard stable population of tsetse confined to the best habitat, i.e., beside a large river.** (DOCX)

**S1 Table. Data for the standard stable population of tsetse confined to the best habitat, i.e., beside a large river.** The indicated timing of reproduction applies to tsetse in all habitats. Effects on death rates of adults and pupae in other habitats are detailed in the main text. (DOCX)

**S2 Table. Initial stable population of male plus female tsetse per cell before any control in the five 13km lengths of river studied in Phase 1, the average percentage of the initial population remaining in the last six months of control, and the average percent remaining in the upstream half of the 13km length, expressed as a proportion of percent remaining in the downstream half, in a number of runs of the model involving increased or decreased values of one parameter in turn, in the context of standard values for all other parameters.** (DOCX)

**S3 Table. Simulated average percent of males plus females remaining in the last six months of control in the five 7km-long plots treated in Phase 1, when all of the standard parameters applied, except for a single change involving either the use of the low kill rates of Schedule 0.01 or 0.02, or the high degradation rates of 4.5% or 5.5% per day.** Figures in parentheses indicate the range of the percents remaining along the individual 1km-long sections of the treated plots. The locations of the plots are indicated in Fig 1B of the main part of the paper. (DOCX)

**S1 Text. Quantifying the changing abundance of tsetse along a river.** (DOCX)

**S1 Original Data. Catches of tsetse from traps located along the Kochi river, Koboko.** (XLSX)

## Author Contributions

**Conceptualization:** Glyn A. Vale, Steve J. Torr.

**Data curation:** Glyn A. Vale, Steve J. Torr.

**Formal analysis:** Glyn A. Vale, John W. Hargrove, Steve J. Torr.

**Funding acquisition:** Andrew Hope, Steve J. Torr.

**Investigation:** Glyn A. Vale, John W. Hargrove, Andrew Hope, Steve J. Torr.

**Methodology:** Glyn A. Vale, Steve J. Torr.

**Project administration:** Andrew Hope, Steve J. Torr.

**Resources:** Steve J. Torr.

**Software:** Glyn A. Vale.

**Supervision:** Steve J. Torr.

**Validation:** Glyn A. Vale, John W. Hargrove, Andrew Hope, Steve J. Torr.

**Visualization:** Glyn A. Vale, Steve J. Torr.

**Writing – original draft:** Glyn A. Vale.

**Writing – review & editing:** John W. Hargrove, Andrew Hope, Steve J. Torr.

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
