## [Decision Letter · Decision Letter 0]

5 Mar 2024

Dear Dr. Torr,

Thank you very much for submitting your manuscript "Modelled impact of Tiny Targets on the distribution and abundance of riverine tsetse" for consideration at PLOS Neglected Tropical Diseases. As with all papers reviewed by the journal, your manuscript was reviewed by members of the editorial board and by several independent reviewers. The reviewers appreciated the attention to an important topic. Based on the reviews, we are likely to accept this manuscript for publication, providing that you modify the manuscript according to the review recommendations. 

This paper present a model of tsetse fly movement, survival and mortality during a seven year control operation. 

The model is based on actual field data, which adds great value to the model's interpretation and its implications. 

 It is a useful training and planning tool for trypanosomosis management. 

There are only some minor improvements suggested by the reviewers, but the only required improvement is to define the objectives more clearly (perhaps even bullet point description).

Sincerely,

Johan Esterhuizen

Guest Editor

Audrey Lenhart

Section Editor

This paper present a model of tsetse fly movement, survival and mortality during a seven year control operation. 

The model is based on actual field data, which adds great value to the model's interpretation and its implications. 

 It is a useful training and planning tool for trypanosomosis management. 

There are only some minor improvements suggested by the reviewers, but the only required improvement is to define the objectives more clearly (perhaps even bullet point description).

Reviewer's Responses to Questions

**Key Review Criteria Required for Acceptance?**

**Methods**

-Are the objectives of the study clearly articulated with a clear testable hypothesis stated?

-Is the study design appropriate to address the stated objectives?

-Is the population clearly described and appropriate for the hypothesis being tested?

-Is the sample size sufficient to ensure adequate power to address the hypothesis being tested?

-Were correct statistical analysis used to support conclusions?

-Are there concerns about ethical or regulatory requirements being met?

Reviewer #1: Methods are well presented.

Reviewer #2: The study is about application of models in determining population dynamics of the riverine tsetse fly. Unfortunately the script does not specify any objectives for the study. These need to be stated clearly as a basis for aligning the study design and its appropriateness. There is no hypothesis stated or being tested. Hence hard to determine if the sample size is sufficient to address the hypothesis test.

Reviewer #3: Interesting manuscript on modeling the impact of targets on tsetse fly.

Methods clearly describe the model, the control and the outputs.

One 'issue' detected in the outputs that should be corrected. Line 333: remove extra bracket i.e [15]]

**Results**

-Does the analysis presented match the analysis plan?

-Are the results clearly and completely presented?

-Are the figures (Tables, Images) of sufficient quality for clarity?

Reviewer #1: Results are well presented, and all figures are fitting

Reviewer #2: The analysis is dependent on several caveats (assumptions or limitations). These do make the model simplistic. Thus the results are based on these caveats. Under this circumstance the results are well discussed. The figures are well presented and of sufficient quality although not easy to interpret.

Reviewer #3: Results clearly and well presented with clear figures.

**Conclusions**

-Are the conclusions supported by the data presented?

-Are the limitations of analysis clearly described?

-Do the authors discuss how these data can be helpful to advance our understanding of the topic under study?

-Is public health relevance addressed?

Reviewer #1: (No Response)

Reviewer #2: There is no specific section gazetted as CONCLUSION. It is hard to tell what the conclusions were made. The script instead ends with THEORETICAL CONSIDERATIONS.

Reviewer #3: Conclusions supported by the analysis and the limitations of the model is discussed.

**Editorial and Data Presentation Modifications?**

Reviewer #1: (No Response)

Reviewer #2: Minor revision recommended. This should proper allow structuring of the paper to accommodate objectives, hypothesis and proper conclusion.

Reviewer #3: Recommend Accept 

Only editorial suggestion is correcting line 333 to remove extra bracket i.e [15]]

**Summary and General Comments**

Reviewer #1: I made a few minor observations that may offer improvement for publication:

Line 124, there was some rogue highlighting on the web address.

Line 134, N-S watershed may not be intuative to an outsider, recomend updating to 'watershed running north to south' or similar

Line 138, Close quaters repitition with 'of the' suggest the 2nd one changes to 'with in'

Line 333. A double bracket is after [15]]

Reviewer #2: This work is good and very important for vector control operations and monitoring. The methodology is novel and only needs to be strengthened by including the element or factor of seasonality. Input data (i.e wet and dry season) need to be segregated.

Reviewer #3: Congratulations on an excellent manuscript on the development of an improved model to test the impact of targets on riverine tsetse fly distribution.

PLOS authors have the option to publish the peer review history of their article (what does this mean?). If published, this will include your full peer review and any attached files.

Reviewer #1: No

Reviewer #2: No

Reviewer #3: No

Figure Files:

Data Requirements:

Reproducibility:

References

---

## [Editor Report · Decision Letter 1]

1 Apr 2024

Dear Dr. Torr,

We are pleased to inform you that your manuscript 'Modelled impact of Tiny Targets on the distribution and abundance of riverine tsetse' has been provisionally accepted for publication in PLOS Neglected Tropical Diseases.

Best regards,

Johan Esterhuizen

Guest Editor

Audrey Lenhart

Section Editor

---

## [Editor Report · Acceptance letter]

6 Apr 2024

Dear Dr. Torr,

We are delighted to inform you that your manuscript, "Modelled impact of Tiny Targets on the distribution and abundance of riverine tsetse," has been formally accepted for publication in PLOS Neglected Tropical Diseases.

Best regards,

Shaden Kamhawi

co-Editor-in-Chief

Paul Brindley

co-Editor-in-Chief
